# SMAP is a pipeline for sample matching in proteogenomics

Ling Li [1], Mingming Niu [2], Alyssa Erickson [1], Jie Luo[3], Kincaid Rowbotham[1], Kai Guo[4], He Huang[1], Yuxin Li [2], Yi Jiang [5], Junguk Hur [6], Chunyu Liu[7], Junmin Peng [2✉] & Xusheng Wang [1✉]

The integration of genomics and proteomics data (proteogenomics) holds the promise of furthering the in-depth understanding of human disease. However, sample mix-up is a pervasive problem in proteogenomics because of the complexity of sample processing. Here, we present a pipeline for Sample Matching in Proteogenomics (SMAP) to verify sample identity and ensure data integrity. SMAP infers sample-dependent protein-coding variants from quantitative mass spectrometry (MS), and aligns the MS-based proteomic samples with genomic samples by two discriminant scores. Theoretical analysis with simulated data indicates that SMAP is capable of uniquely matching proteomic and genomic samples when ≥20% genotypes of individual samples are available. When SMAP was applied to a large-scale dataset generated by the PsychENCODE BrainGVEX project, 54 samples (19%) were corrected. The correction was further confirmed by ribosome profiling and chromatin sequencing (ATAC-seq) data from the same set of samples. Our results demonstrate that SMAP is an effective tool for sample verification in a large-scale MS-based proteogenomics study. SMAP is publicly available at https://github.com/UND-Wanglab/SMAP, and a web-based version can be accessed at https://smap.shinyapps.io/smap/.

[1] Department of Biology, University of North Dakota, Grand Forks, ND 58202, USA. [2] Departments of Structural Biology and Developmental Neurobiology, Center for Proteomics and Metabolomics, St. Jude Children's Research Hospital, Memphis, TN 38105, USA. [3] State Key Laboratory for Managing Biotic and Chemical Threats to the Quality and Safety of Agro-products, Zhejiang Academy of Agricultural Sciences, Hangzhou 310021, China. [4] Department of Neurology, University of Michigan, Ann Arbor, MI 48109, USA. [5] Department of Epidemiology and Biostatistics, School of Public Health, Tongji Medical College, Huazhong University of Science and Technology, Wuhan 430030, China. [6] Department of Biomedical Sciences, School of medicine and health sciences, University of North Dakota, Grand Forks, ND 58202, USA. [7] Department of Psychiatry, SUNY Upstate Medical University, Syracuse, NY 13210, USA. ✉email: junmin.peng@stjude.org; xusheng.wang@und.edu

With the development of high-throughput technologies in recent years, remarkable accomplishments have been made by well-conceived and large-scale projects carried out by large consortia, such as the Cancer Genome Atlas (TCGA) project[1], the Clinical Proteomic Tumor Analysis Consortium (CPTAC) project[2–5], the Genotype-Tissue Expression (GTEx) project[6], and the Encyclopedia Of DNA Elements (ENCODE) Project[7]. These studies usually involve collecting single- or multi-layer omic data from a large number of subjects, followed by statistical significance tests. The power and effectiveness of the statistical tests rely on the accuracy of sample identity[8]. However, sample mix-up in large-scale multi-omics studies, as a result of specimen labeling errors, or data mismanagement, is a widespread problem[9]. Such error is often neglected, and it generally leads to irreproducible results, weakened statistical testing power, and false conclusions[10]. Therefore, to reduce the bias and increase the precision for subsequent analyses, verification of sample identity is one of the first steps in a large-scale omics project.

Mass spectrometry (MS)-based proteomics has emerged as an important molecular profiling technology[11] that is complementary to other omics, such as genomics and transcriptomics. Due to recent technical advances, MS-based proteomics has undergone rapid development yielding numerous large-scale proteomic data[2–5]. Most of these data are generated by multiplexed isobaric labeling-based protein quantification methods, such as isobaric tags for relative or absolute quantitation (iTRAQ) and tandem mass tag (TMT). For example, the TMT-based strategy can now measure as many as 27 samples in one batch simultaneously[12], and potentially allows for the measurement of 81 samples when combined with metabolic labeling by amino acids in cell culture (SILAC). These multiplexing strategies would further exacerbate the problem of sample mix-up, posing a significant challenge for sample verification and calibration.

Several methods have been developed to verify sample identity in large-scale genomic and transcriptomic studies using genotype concordance[13], correlation of mRNA and protein[14], and correlation of variant fractions[15]. For example, we have also implemented a genotype-based method to address the sample mix-up problem from sequencing-based transcriptomic data[16]. Most of these methods exploit the genotypic information of a sample, which can be directly derived from sequencing data. Calling genotype from multiplexed isobaric labeling-based quantitative proteomic data remains a major challenge because multiple labeled samples are mixed during the experiment. While the proteogenomics approach has been widely used to detect variant peptides, it only calls variant peptides from label-free or isobaric labeled proteomic data[17–19]. As a result, there is no method available for verifying and calibrating sample identity in a multiplexed quantitative proteomic study.

In this study, we present a pipeline for Sample Matching in Proteogenomics (SMAP). SMAP first performs proteogenomics to detect variant peptides from multiplexed isobaric labeling-based quantitative proteomic data, and then infers genotypic information of each sample based on the expression level of the variant peptides. SMAP finally verifies and calibrates sample identity based on a combination of concordance and specificity scores using inferred genotypic information. The performance of SMAP is assessed by a simulation study and a large-scale proteomic dataset.

## Results

**Method implementation**. SMAP consists of three main components (Fig. 1a and Supplementary Fig. 1): (i) identifying variant peptides using the proteogenomics approach; (ii) inferring genotypic information for each sample; and (iii) verifying and calibrating sample identity. SMAP can take a quantification table of variant peptides from a proteogenomics analysis. In this study, variant peptides are identified by JUMPg[19], which constructs a customized database using genomic variant files (e.g., VCF files) or indirectly from RNAseq raw data in FASTQ format, in which JUMP performs preprocessing, tag generation, MS/MS pattern matching, and scoring as previously reported[20]. The identified variant peptides are further filtered with the target-decoy strategy to control the false discovery rate (FDR)[21,22].

SMAP infers sample genotypic information based on the relative expression level of variant peptides identified in proteomic data with multiplexed isobaric labeling-based quantification methods (Fig. 1b). For each spectrum, the intensity of reporter ions is extracted as the expression level for each sample (Fig. 1c). The intensity in each sample is transformed to log$_2$-scale, followed by a scale normalization using the following formula,

$$y_i = \frac{x_i - \min(x)}{\max(x) - \min(x)} \tag{1}$$

Where $x$ is the intensity across all samples; $x_i$ and $y_i$ are raw and scaled intensity for a specific sample, respectively. The scaled intensities are in the range of 0 to 1.

SMAP then uses the proteogenomics approach to detect variant peptides (Fig. 1d), followed by inferring genotypic information of each sample based on the scaled intensity of identified variant peptides (Fig. 1e). For a diploid organism, such as human or mouse, there are three possible genotypes of a variant peptide: nonmutant homozygous genotype (AA), mutant homozygous genotype (BB), and mutant heterozygous genotype (AB). To determine the genotypic information of each sample of a variant peptide, SMAP divides the scaled intensities into three quartiles: lower-quartile (LQ, 25th percentile), interquartile (IQ, 25th to 75th percentiles), and upper-quartile (UQ, 75th percentile) (Supplementary Fig. 2). The quartiles are assumed to correspond to the nonmutant (AA), heterozygous (AB), and mutant (BB) genotypes, respectively. We also used the genotype dosage information in the genotypic data as prior knowledge to assign the inferred genotypes (Supplementary Fig. 3). For example, if there are only two genotypes (e.g., mutant and nonmutant) in the genotype file, the program will use a cutoff of 0.5 to assign the two genotypes by excluding the possibility of inferring a heterozygous genotype.

Once the genotypic information of each sample is inferred, SMAP verifies and calibrates the sample identity using a combination of two scores: concordance score (i.e., Cscore) and specificity score (delta concordance score: △Cscore) (Fig. 1a and Supplementary Fig. 1). The Cscore is defined as the percentage of matched genotypes. The △Cscore is defined as the difference between the best concordance score and the following concordance score divided by the best score. Thereby, the △Cscore is a good measure of separating true from false hits. A confident assignment generally has a higher Cscore and △Cscore. To determine whether a sample can be verified, these two scores are combined for calibrating its identity.

**Performance of SMAP using simulation data**. Sample verification and calibration depend largely on reliable genotypic information derived from the quantitative proteomic data. To determine how many reliable genotypes in a sample are sufficient for SMAP to verify and calibrate its identity, we conducted a simulation study using a subset of genotypic data (Fig. 2a). To mimic "real" genotypic information derived from a large-scale proteomic dataset, we first extracted a subset of the genotypic matrix composed of 500 SNPs from 420 samples. We generated a simulated dataset in six steps (Supplementary Fig. 4): (1) estimating the frequency of each genotype across all samples; (2)

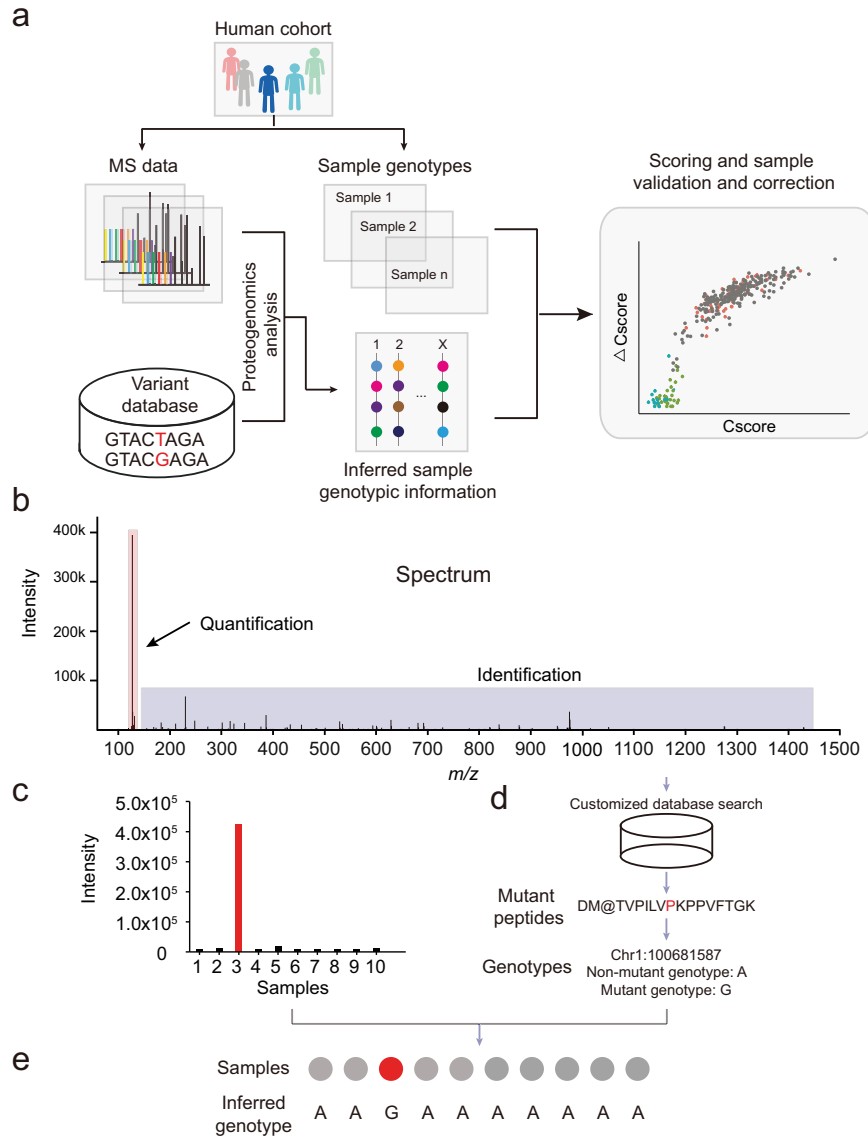

**Fig. 1 Architecture of SMAP. (a)** Schematic diagram of the SMAP pipeline. Mass spectrometry (MS)-based proteomic data are generated from a large-scale project. The data are searched against the customized database containing all theoretical variant peptides generated from whole-genome DNA sequencing, RNA sequencing, or public databases (e.g., dbSNP). For each batch of the TMT-based quantitative proteomic data, variant peptides are identified through proteogenomic analysis. Two scores (i.e., concordance score and delta concordance score) are computed for verifying and calibrating the sample identity. **(b)** Example of an MS spectrum containing TMT report ions. **(c)** A zoom-in view of the intensity of reporter ions is shown in panel **b**. **(d)** Identification of variant peptides using the proteogenomics approach. **(e)** Inferring sample genotypes from quantitative expression level of the variant peptide for each sample.

randomly selecting a sample *i*; (3) randomly selecting a genotype *j* in sample *i*; (4) choosing another genotype with a frequency estimated in step 1; (5) swapping the genotype *j* in sample *i* with a chosen genotype in step 4; and (6) repeating steps 1–5 to generate a simulated dataset with a certain percentage of the error rate (e.g., 10%, 20%, 40%, and 80%). We examined whether the sample identity assigned by SMAP matched the original sample identity based on the combination of Cscore and △Cscore. As shown in Fig. 2b–d, SMAP is capable of successfully validating the sample identity with the reliable genotypic number as low as 20% (i.e., 80% shuffled).

**Application of SMAP to PsychENCODE BrainGVEX proteomic data**. We next applied SMAP to a deep proteomic dataset

generated by the PsychENCODE BrainGVEX project, in which 288 biological samples and 31 internal controls (i.e., a mixture of 288 samples) were quantified by 29 batches of 11-plex TMT-based proteomic technology. In addition to proteomic data, the PsychENCODE BrainGVEX project also generated other omic data with matched samples, including 285 samples with low-depth whole-genome sequencing (WGS), 426 samples with RNA sequencing (RNA-Seq), 295 samples with assay for transposase-accessible chromatin using sequencing (ATAC-Seq), and 197 samples with ribosome sequencing (Ribo-Seq). Previous analysis using the DRAMS program has identified ~19% mis-labeled samples in both WGS and ATAC-seq, ~25% in Ribo-seq, and ~3% in RNA-seq data[16].

To identify variant peptides in the PsychENCODE BrainGVEX proteomic data, approximately 34 million MS2 spectra from 29

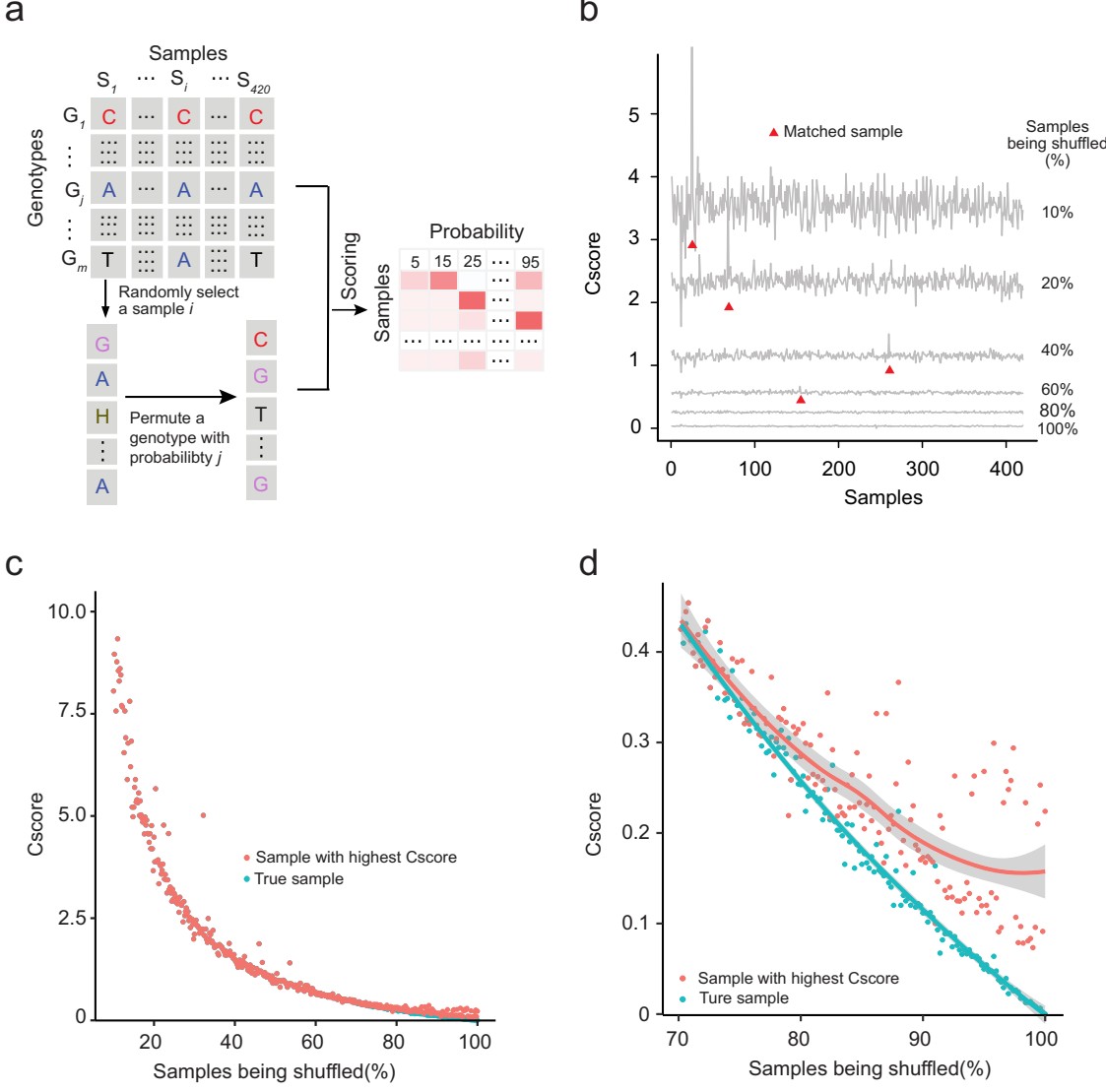

**Fig. 2 Parameterization and evaluation of SMAP performance using a simulation study.** (**a**) Diagram showing the simulation procedure. See Supplementary Fig. 4 for a more detailed procedure. "H" is used to denote heterozygote. (**b**) Curve plot showing Cscore distribution. The X-axis represents 420 samples and the y-axis represents Cscore between a testing sample and each sample in the simulation data. (**c**) Score distribution of the matched sample with the highest Cscore and "true" sample along with the percentage of samples being shuffled. The X-axis represents the percentage of samples being shuffled and the Y-axis denotes Cscore. The red dot is the "true" sample and the blue is the one with the highest score. (**d**) A zoom-in plot of the score distribution with samples being shuffled from 70% to 100%. Smoothing curves based on linear model separately for each group are shown in red and blue, respectively, with 95% confidence intervals in grey.

batches of 11-plex TMT experiments were searched against a customized database. The database contains 20,396 reviewed protein sequences from the UniProt database[23] and 17,844,001 theoretical peptides translated from variants generated by WGS and RNA-seq data. SMAP identified a total of 5,065 unique variant peptides (Supplementary Data 1), corresponding to 20,129 SNPs at 5% peptide FDR. The number of variant peptides and SNPs show similar trends across 29 batches. On average, a total of 694 SNPs was identified per batch, ranging from 430 SNPs in batch 23 to 901 SNPs in batch 24. To use reliable SNPs, further filtering of SNPs was completed using the minor allele frequency (MAF) of >1%, identifying a total of 8,358 SNPs with an average of 288 SNPs per batch (Fig. 3a).

To determine genotypic information of a sample for each identified variant peptide, two parameters are critical: a threshold of the noise signal and a signal-to-noise (S/N) ratio (Fig. 3b). For a

variant peptide, the signal of a sample with a nonmutant homozygous genotype (AA) is expected to be no different from the background noise because the variant peptide is not detectable in the sample. To determine the threshold of the background noise level, we used one batch (i.e., Batch 1) of the 29 TMT experiments to determine the distribution of the background noise level of all identified variant peptides. We found that the distribution can be clearly separated into two modes by the expectation-maximum (EM) method, consisting of an authentic noise distribution (mean = 14.06, SD = 1.39; $Log_2$ intensity) and a signal distribution (mean = 18.22, SD = 1.29; $Log_2$ intensity) (Fig. 3c). The result indicates that a $log_2$ intensity of 16.14 (noise mean + 1.5 SD or signal mean − 1.5 SD) can be used as a cutoff. This is consistent with the result found when evaluating the impact of the minimal intensity on the number of genotypes and the Cscore (Fig. 3d). The analysis of the S/N ratio between the mutant homozygous

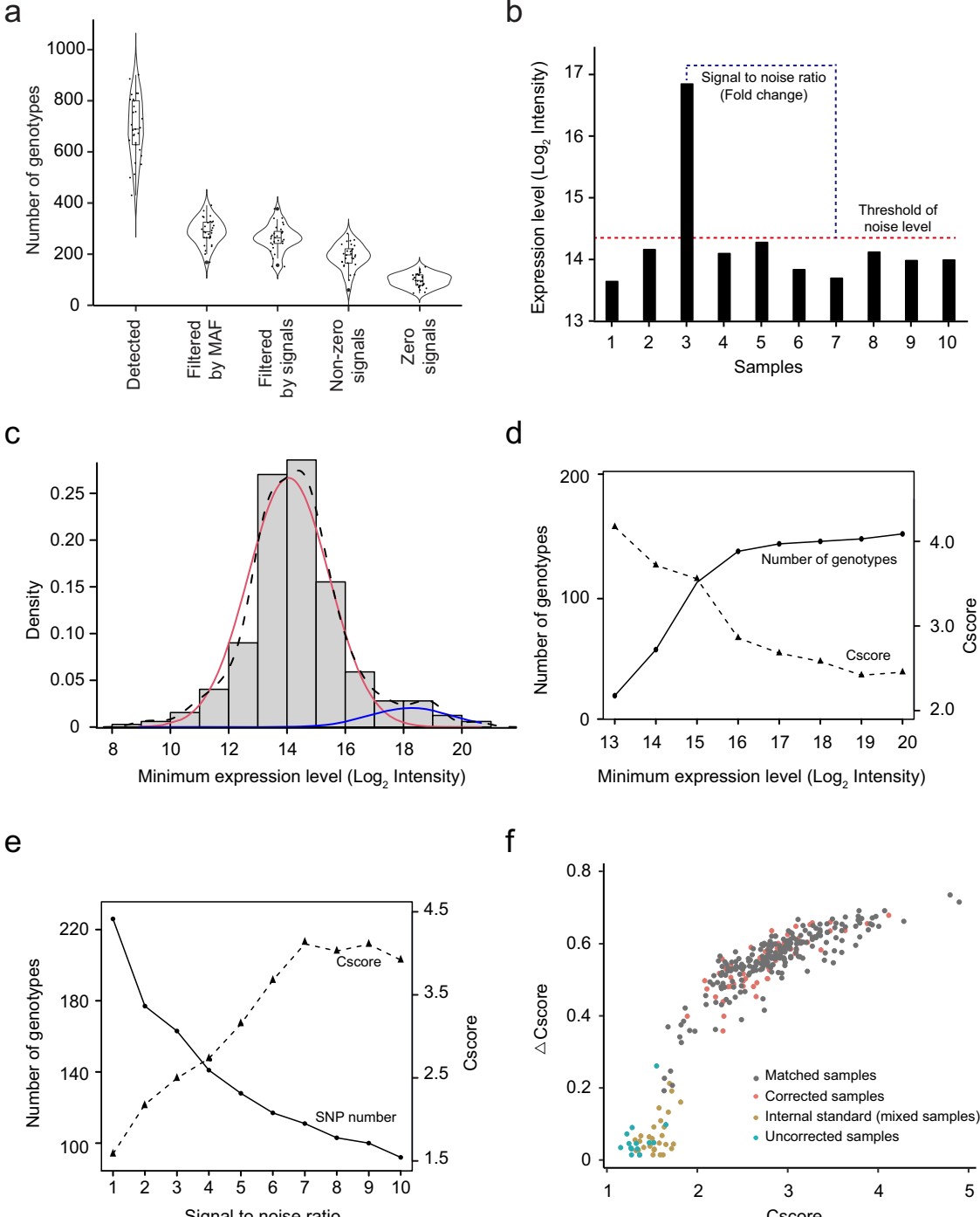

**Fig. 3 Application of SMAP to the PsychENCODE BrainGVEX proteomic data.** (**a**) Distribution of the number of SNPs identified, filtered by both minor allele frequency (MAF), threshold of the noise signal, and identified variant peptides with and without zero signals. The numbers of genotypes from 29 batches are shown as violin and box-and-whisker plots. The box-and-whisker plot indicates the median (line within box), 25–75% quartile range (box), and 10–90% range (whiskers). (**b**) Diagram showing how to define the threshold of the noise background and the signal-to-noise (S/N) ratio. (**c**) Distribution of minimum signals of identified variant peptides. The minimum expression can be separated into noise and signal distributions by the expectation-maximum (EM) method: all values (black dash line), noise values (red solid line), and signal values (blue solid line). (**d**) The distributions of the number of genotypes and Cscore with different thresholds of the minimum expression level. (**e**) The distributions of the number of genotypes and Cscore with different S/N ratios. (**f**) Score distribution of matched samples, corrected samples, internal controls, and uncorrected samples across all 29 batches.

genotype (BB) and the nonmutant homozygous genotype (AA) showed that an S/N value of 3 achieves a reasonably good number of genotypes and Cscore (Fig. 3e). Following filtration by the minimum signal and the S/N ratio, we detected 7628 genotypes, with an average of 263 genotypes per batch (Fig. 3a).

After optimizing the parameters, we used 7628 filtered SNPs to infer genotypic information for 288 biological samples and 31 controls in the PsychENCODE BrainGVEX proteomic data (Supplementary Data 2). The genotypic information of each sample was then inferred based on its expression level in a batch

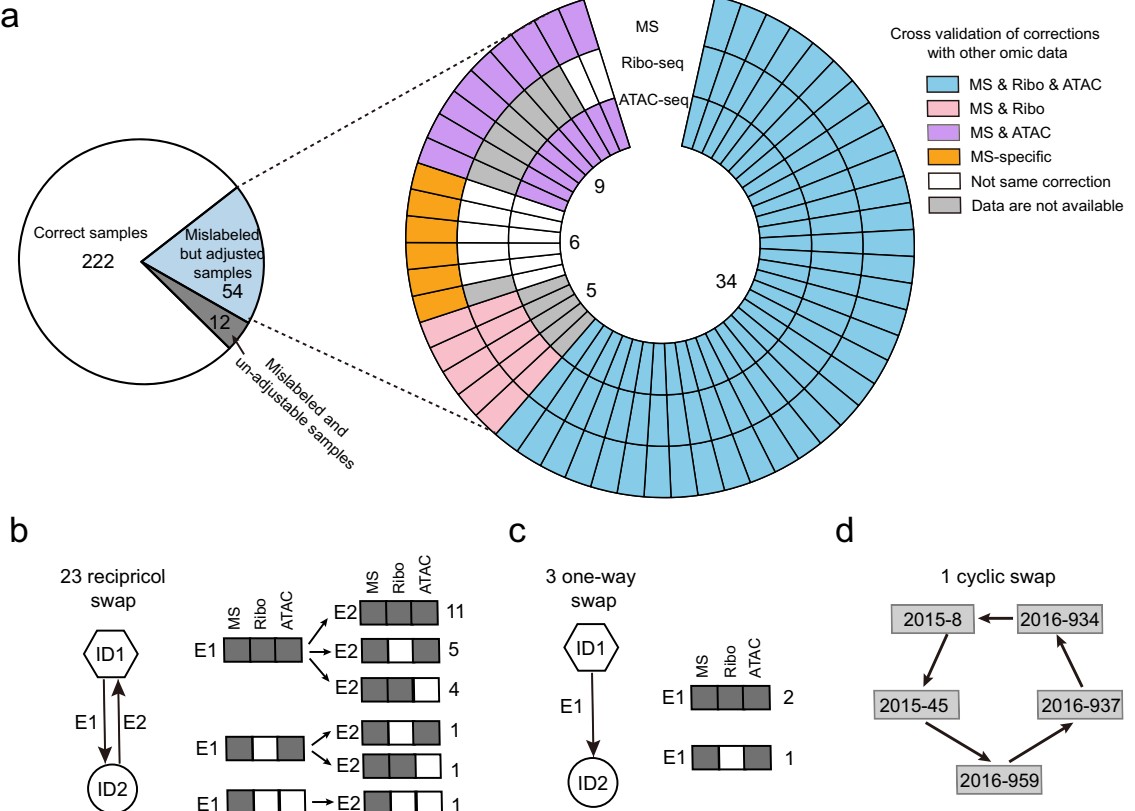

**Fig. 4 Validation of sample calibration across other omic platforms.** (**a**) The pie diagram shows the number of correct samples, mislabeled but adjustable samples, and mislabeled and un-adjustable samples. The circular plot shows the validation of sample corrections across different platforms. Each circle represents an omic platform. (**b–d**) Three types of sample calibrations and comparison of calibrations between different omic platforms.

using the strategy of three quartiles (Supplementary Fig. 3). To verify the sample identity, we first evaluated the distribution of Cscore and △Cscore of the 31 internal controls, which were mixed samples that function as a negative control. The internal controls showed an average Cscore of 1.50 and a standard deviation of 0.13 (Supplementary Fig. 5a), and an average △Cscore of 0.05 and a standard deviation of 0.05. When applying SMAP to the first batch of the proteomic data, the samples can be clearly separated from internal controls and matched samples with the Cscore of 1.50 and △Cscore of 0.20, identifying one sample as being mixed up (Supplementary Fig. 5b). For 288 biological samples, SMAP identified 276 biological samples with the Cscore above 1.50 and △Cscore above 0.20. When comparing those 276 samples with their original labeling identities from DNA-based genotype data, we found that 54 samples (18.75%) showed mixed-up identification (Fig. 3f). In addition, a total of 12 samples showed a combined score below the threshold, suggesting that these samples could not be calibrated for their identities.

**Cross-validation of sample correction.** The multi-omic data generated by the PsychENCODE BrainGVEX project enable cross-validation of sample calibration made by SMAP. As the identity of samples has previously been verified and calibrated in other omic data, such as samples used for ATAC-seq and Ribo-seq data[16], we next evaluated whether sample identities in the proteomic data calibrated by SMAP are consistent with those made in both ATAC-seq and Ribo-seq data. We found that 34 out of 54 samples (63%) showed the same calibration in samples from all three platforms: MS-based proteomic, ATAC-seq, and Ribo-seq. In addition, a total of 9 samples had the same calibration between

proteomics and ATAC-seq, 5 between proteomic and Ribo-seq, and 6 proteomic-specific calibrations (Fig. 4a).

We next sought to investigate how these 54 samples were mixed in the proteomic experiment. We summarized the calibrations into three categories: reciprocal, one-way, and cyclic swaps. We found that 46 samples (23 pairs) were reciprocally swapped (Fig. 4b), with about half of the samples (11 pairs) supported by the same calibration in all three platforms: MS-based proteomic, ATAC-seq, and Ribo-seq, followed by nine pairs that were supported by all three platforms in one direction but by two platforms in the other direction. For the three one-way swaps, two calibrations were supported by all three platforms and one by two platforms (Fig. 4c). Most of the cyclic swaps (5/6) were proteomic-specific calibrations (Fig. 4d). The result suggests that sample mix-up is not a random process rather follows a certain pattern in the lab procedures.

**Performance evaluation of SMAP compared to COSMO.** To assess the performance of SMAP, we also analyzed our data using COSMO[14], a program recently developed to correct sample mislabeling in omic data through correlating mRNA and protein expression levels. A key step in the COSMO is feature selection by selecting highly correlated gene and protein pairs. With the default correlation coefficient cutoff of 0.5 in COSMO, only a few features were selected in our data probably due to about 20% of the samples being mislabeled based on the correction made by SMAP. To obtain an optimal number of features, we examined correlation coefficients ranging from 0.2 to 0.55, and found that the correlation coefficient of 0.35 produced the best performance when considering the number of selected features and matched samples (Supplementary Figs. 6a, b).

With the selected correlation coefficient of 0.35, COSMO detected 68 mislabeled samples out of a total of 288 samples in our proteomic data. We found that SMAP and COSMO made the same correction for 270 (94%) samples, including 218 samples matched to its original ID, and 52 mislabeled samples but corrected to the same identity by both methods. In the remaining 18 (6%) samples that showed inconsistent correction between COSMO and SMAP (Supplementary Fig. 6c), we observed that 4 samples matched to the original identity by SMAP were "mis-assigned" to other identities by COSMO. For example, SMAP made no change for the sample *S2015_1504*, with a high Cscore and ΔCscore (Supplementary Fig. 6d); however, it was re-assigned to a new sample identity by COSMO. Conversely, SMAP also "mis-assigned" 2 samples, but COSMO indicated that the samples had high correlations. For example, the sample *S2015_1477*, showed a significant correlation of the expression levels between protein and RNA with the original sample ($r = 0.65$, $p = 1.9 \times 10^{-14}$) (Supplementary Fig. 6e). Two samples were adjusted into different sample identities by both SMAP and COSMO with high scores. In addition, we found that 10 samples cannot be corrected by both SMAP and COSMO because of a low average Cscore of $1.30 \pm 0.12$ ($2.83 \pm 0.55$, on average, from 218 matched samples) and a low average correction coefficient of $0.10 \pm 0.04$ ($0.45 \pm 0.09$, on average, from 218 matched samples), presumably due to no matched samples between proteomic and transcriptomic data.

## Discussion

Sample mix-up is a pervasive problem in large-scale omic studies. In this study, we described a method, SMAP, for validating and correcting samples that are used for a large-scale TMT-based quantitative proteomic study. By applying SMAP to proteomic data from 288 biological samples in the pyschENcode BrainG-VEX project, we verified and calibrated ~18% mislabeled samples. Based on the simulation study, SMAP has proven capable of validating the sample identity of as few as 20% of reliable genotypes.

While the proteogenomics approach is increasingly used to detect protein-coding variants and mutant splicing sites[17–19], there is no study to infer genotypic information for each sample in multiplexed isobaric quantitative proteomic data. It is the first time that SMAP determines genotypic information of each sample based on the expression level of the variant peptides. With advances in next-generation sequencing and high-resolution mass spectrometry technologies, omic data from the same set of biological samples are now routinely collected in a project. Appropriate data quality controls at each step are required to ensure high data integrity. Therefore, SMAP complements existing approaches for sample verification and calibration that have been developed for genomic and transcriptomic data.

Correction of sample mislabeling is paramount to all large-scale omic studies. The SMAP program centers on sample correction for proteomic data by leveraging genotypic data. This concept of using genotypic data for sample correction can be applied to other omic data. The genotypes of each sample can be readily extracted from all massive sequencing data, ranging from whole-genome DNA sequencing, epigenome (e.g., ATAC-seq), to the transcriptome (bulk RNA and single-cell RNAseq). However, inferring genotypes from MS data is a much greater challenge than from sequencing data as we need to convert MS signals into amino acids and then convert amino acids into genotypes. More importantly, we proposed a scoring scheme in which a combined score is generated from two scores: (i) Cscore, which is a measure of the goodness of fit of genotypes inferred from proteomic data to non-mutant genotypes; (ii) ΔCscore: is a measure of the specificity of the fit. This new scoring scheme can be adapted to any other algorithms for sample correction.

Although SMAP can infer sample-level genotypic information for sample verification, the inference certainty was still imperfect. First, the expression level of a variant peptide is quantitative, unlike variants called from genomic and transcriptomic data generated by sequencing technologies. Although we proposed to use three quartiles to determine the genotypes, the genotypes inferred by SMAP show uncertainties. In the analysis of the PsychENCODE BrainGVEX proteomic data, we found many mismatched genotypes were in the boundary of each range. Second, the homozygous mutant genotype (BB) and heterozygous genotype (AB) cannot be distinguished solely depending on the expression level if only AB genotypes are present in the samples. To mitigate this issue, SMAP examines the number of genotypes in the original genotypic data (Supplementary Fig. 3). Third, expression imbalance, such as single-parental expression (e.g., imprinted expression) and over- or under-dominant expression of the identified variant peptides would potentially influence the inference of genotypic information[24].

Large-scale omic data profiling and analyses typically involve many research laboratories and mistakes could occur in different ways. Sample mix-up is not only a common problem but also a major problem getting less attention than it deserves. By comparing data of multi-omic data, we had the best chance of correct sample identifies, which is essential for performing high-quality research. In summary, we present a robust and easy-to-use method and tool, SMAP, for sample verification and calibration. SMAP successfully calibrated ~18% mixed samples in a large-scale proteomic dataset, demonstrating that our scoring scheme by combining Cscore and ΔCscore is effective in sample verification. We recommend sample verification and calibration as an important part of the data analysis in a large-scale MS-based proteomic study.

## Methods

**Proteomic and other omic datasets**. A total of 288 well-characterized postmortem human brain samples (179 males and 109 females) from the Stanley Medical Research Institute and Banner Sun Health Research Institute were used for this study[25]. These samples were collected from 210 neurotypical controls, 49 individuals with schizophrenia (SCZ), and 29 individuals with bipolar disorder (BD). The samples include 282 Caucasians, 1 Hispanic, 1 African American, 3 Asian American, and 1 unknown. The brain frontal cortex samples were subjected to 29 batches of deep proteome profiling using TMT LC/LC-MS/MS technology. In addition to the proteomic data that we generated, five other omic data are also available at the psychENCODE knowledge portal (https://psychencode.synapse.org/Explore/Studies/DetailsPage?studyName=BrainGVEX), including RNA-Seq data from 426 samples (274 males and 152 females), assay for transposase-accessible chromatin using sequencing (ATAC-Seq) data from 295 samples (180 males, 112 females, and 3 unknown-sex samples), and ribosome sequencing (Ribo-Seq) data from 197 samples (122 males, 70 females, and 5 unknown-sex samples). The genotypes were generated by the combination of three platforms, including Affymetrix chip data, whole-genome sequencing, and RNA-seq data[16].

**Identification of variant peptides in the proteomic data**. JUMPg program was used to identify variant peptides by taking MS-based proteomic data and a customized database curated with genomic variant data[19]. In JUMPg program, the variant peptides were identified by the JUMP search engine, which performs preprocessing, tag generation, MS/MS pattern matching, and scoring[20]. JUMP was used to search MS/MS raw data against a composite target/decoy database[21,22] to evaluate the false discovery rate (FDR). In JUMPg, a two-stage FDR method is used: in the first stage, the MS/MS data are searched against a reference protein database, and the confidently identified spectra are removed; in the second stage, the remaining spectra are searched against the variant protein database, and the FDR for variant peptides is calculated based on the second stage search results. The target database contains both core protein sequences downloaded from UniProt[23] and all theoretical variant peptides generated from all non-synonymous protein-coding variants. To generate theoretical variant peptides, all variants in the genotypes were re-annotated using the genome annotation tool ANNOVAR[26] based on the human reference genome (i.e., GRCh37/hg19 assembly). We used the UCSC gene annotation model. The decoy database was generated by reversing protein sequences in the target database. Major parameters included precursor and product ion mass tolerance (±8 ppm), full trypticity, static mass shift for the TMT

tags (+229.16293) and carbamidomethyl modification of 57.02146 on cysteine, dynamic mass shift for Met oxidation (+15.99491), maximal missed cleavage ($n = 2$), and maximal modification sites ($n = 3$). To filter variant peptides, PSMs were first filtered by user-specified parameters (e.g., minimum peptide length and minimum search score), then by precursor ion mass accuracy. The resulting PSMs were further grouped by precursor ion charge state and tryptic ends and then filtered by matching scores (Jscore and ΔJscore) to achieve a peptide FDR < 5%. If one peptide could be generated from multiple homologous proteins, the peptide was assigned to the protein with the highest PSM based on the rule of parsimony.

**TMT-based Peptide/Protein Quantification by JUMP Software Suite.** The analysis was performed in the following steps[27]: (1) extracting TMT reporter ion intensities of each PSM; (2) correcting the raw intensities based on the isotopic distribution of each labeling reagent (e.g., TMT126 generates 91.8%, 7.9%, and 0.3% of 126, 127, 128 m/z ions, respectively); (3) removing sample loading bias by normalization with the trimmed median intensity of all PSMs; (4) calculating the mean-centered intensities across samples (e.g. relative intensities between each sample and the mean), (5) summarizing relative intensities of proteins or variant peptides by averaging related PSMs. The quantification values of variant peptides with and without missing values were extracted for inferring genotypic information for each sample by SMAP.

**Simulation data.** We generated a simulated dataset to test the performance of SMAP by evaluating how many genotypes identified from MS-based proteomic data are needed for validating and correcting the sample identity. We first randomly extracted a subset of a matrix with 500 genotypes and 420 samples. We then randomly selected a sample from the matrix and permuted the genotypic information with replacement sampling at a certain probability level α (Supplementary Fig. 4). The α level was set in the range from 10 to 100 at a step of 1.

**Reporting summary.** Further information on research design is available in the Nature Research Reporting Summary linked to this article.

## Data availability
The mass spectrometry data associated with this study are provided in Supplementary Data 1 and Supplementary Data 2. The raw mass spectrometry data used in this study are available in the Synapse database under accession code syn26231732.

## Code availability
The SMAP is implemented in both standalone (https://github.com/UND-Wanglab/SMAP) and web-based (https://smap.shinyapps.io/smap/) versions. The standalone SMAP is written in Perl programming language. The web-based SMAP is developed using Shiny, an R package that supports the development of web-based R applications that can be hosted online. Manual, tutorial, and sample datasets are also available at the SMAP project homepage https://sites.google.com/view/smapwanglab/home.

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

## Acknowledgements
This work was partially supported by two NIH pilot grants to X.W.: 1P30DA044223 (PI: Robert Williams), 5P20GM104360 (PI: Roxanne Vaughan). It is also supported by the UND Vice President for Research & Economic Development (VPRED) seed program (X.W.). The MS analysis was performed in the Center of Proteomics and Metabolomics at St. Jude Children's Research Hospital, partially supported by two NIH Grants to J.P.: P30CA021765 and R01AG053987, and three grants to C.L.: U01MH103340-01, 1U01MH116489, and 1R01MH110920.

## Author contributions
X.W. and J.P contributed to the conception and design of the project. X.W., L.L., A.E., J.L., K.R. and Y.L. developed computational algorithms and wrote the SMAP program. K.G., Y.J., H.H. and J.H. performed data analysis. M.N. performed proteomic experiments. C.L. provided human specimens and genotype data. X.W. and L.L. wrote the manuscript.

## Competing interests
The authors declare no competing interests.
