## [Peer Review File · Nature Communications]

SMAP is a pipeline for sample matching in proteogenomicsReviewers' Comments:

Reviewer #1:

Remarks to the Author:

In this manuscript, the authors present an analysis pipeline for determining if there are sample mix-ups during proteogenomics analysis. They demonstrate that through correlation of variant-containing peptides, one can correct for sample mix-ups with high accuracy.

The paper was clearly written, easy to follow, and the codebase and documentation seemed to be well put together.

However, I have two concerns.

First, unfortunately, there has been a recent publication in *Patterns* on a similar idea (PMID: 34036290). In that paper, the authors demonstrate the idea of sample-MS dataset matching with a broader scope, including several consortium datasets and participants. At the least, now, there would need to be a comparison of the SMAP approach against their COSMOS approach.

Second, and unrelated to the first comment, I feel that the scope of the work is quite focused on one problem (sample mix-ups in proteomics) and would need to be significantly broadened for this journal.

Reviewer #2:

Remarks to the Author:

The authors present a method to identify samples with mismatched labels in multi-omics studies, where both proteomics and genomic/genetic datasets are available. Specifically, the authors build upon their previous work (the JUMPg framework, which in turn builds on the JUMP database search algorithm). The method importantly fills the gap in current methods that address isobaric labeling datasets, such as TMT and iTRAQ. It addresses an important concern (sample mismatches) which has been receiving necessary attention recently, by extending the remediation strategies to a new data type.

While the central concepts and strategies can be understood, the authors need to explicitly define many essential terms, clean up several typographical errors and add more detail to the methods.

Please address the following concerns:

-Lines 106-112: Allelic information – Beyond the quantile argument, can't you also determine the intensities of the non-mutant peptides that overlap that region, and compare intensities to determine AA, AB or BB? So, if the mutant and non-mutant peptides occur in nearly equal intensities in the same sample, you have a heterozygous mutation.

-Line 115: Please define the concordance score: it is not the percentage of overlapping genotypes. So how is it defined?

-Lines 129-130: How exactly are the authors shuffling genotypes: are they starting with genotype dosages for the alternate allele, and then randomly mutating to one of the other two values? For example, if for the true sample, SNP *i* has an alternate allele dosage of 2, are they mutating it randomly to 0 or 1? Or just randomly choosing SNP dosages from the other samples? In the latter case, one may randomly choose the same SNP dosage as before (in the above example, SNP *i* dosage = 2 may be replaced with with SNP *i* dosage = 2 from the other samples). This would result in an overestimation of the genotype substitution rate.

Or are they doing something else?

Figure 2 is confusing as it implies that the bases themselves are randomly bent mutated (in both

haplotypes?). Please clarify this, as the exact definition has implications for how the shuffling process would reduce concordance.

-Lines 154-171: The threshold and S/N ratio cutoffs are defined in terms of homozygous alleles only. In the EM method, the authors identify only two peaks. Will this choice of cutoffs produce a bias against the detection of heterozygotes? That will then have an impact on the allele frequencies of the SNPs that can be detected. What is the MAF distribution of detected SNPs?

-Lines 265-283: What are the error rates from the previous pipelines (JUMP, JUMPg)? While a comprehensive comparison is not necessary, could you comment on the rate of occurrence of ambiguous variant peptides in your datasets? How was the filtering carried out? Stating this in the manuscript would be very helpful.

-Supplementary Figure 3: What are the unmatched samples? Are those internal controls? Have all the samples in the first batch been matched?

Minor edits:

-Figure 4a: Label for the orange swatch (4th from the top) is incorrect – should be “MS-specific”.

-Figure 2 caption: Lines 350-351: Maybe change the names of the samples: “selected sample with eh highest score” and “select sample” are too similar. Make the latter “true sample”.

-Figure 2a: What does the H represent in the figure? Heterozygote?

-Lines 32-33: The wording seems to imply that 18.8% of the mismatched samples were found and corrected (and that 71.2% of mismatched samples were not). The actual situation is that 18.8% of all samples were found to be mismatched, and that these were detected using SMAP. Please clarify the language here.

-Line 91: FASTQ file, not FASTAQ file

-Figure 1c: Why is sample 11 shown in red? Are the authors highlighting anything in particular? Maybe Sample 3 should be red instead?

-Line 136: The 31 internal controls are “mixture(s) of the 288 samples”? How are they “mixed”? Please explain.

-Line 142: Mixed samples, or mismatched samples? I think mismatched is intended, so please use consistent terminology.

-Line 149: This reference to Figure 3a should be probably be removed. The numbers just quoted (~20,000 SNPs) do not appear in Figure 3a.

-Figure 3b: The black dotted line should end at the red threshold, right? The ratio is of the BB to AA intensities (with the AA intensity being defined by the red line)?

Prashant Emani

Reviewer #3:
Remarks to the Author:

In this manuscript, Li et al. present a novel pipeline named SMAP that makes use of both genomics data and MS-based proteomics data to identify and correct mislabeled samples in proteogenomic studies. Using multi-omics data in proteogenomics studies for mislabeled sample identification and correction is not new. A published tool called COSMO from the precisionFDA NCI-CPTAC Multi-omics Enabled Sample Mislabeled Correction Challenge has shown the value to combine proteomics data with other omics data including genomics data in mislabeled sample identification and correction (PMID: 34036290). However, the use of variant peptide identification from proteomics in SMAP is somewhat novel. The authors tested SMAP on a large-scale proteogenomic dataset in which the proteomics data contains 29 batches of TMT11-plex data and compared SMAP with the results using other omics data from the same cohort. This pipeline may be useful in proteogenomics studies. However, there are a few major and minor issues needed to be addressed.

Major issues:

1. Variant peptide identification in proteogenomics has been known to be error-prone especially in searching large non-sample specific customized databases in which most sequences in the search space are false. In this study, a single customized database (17,844,001 theoretical peptides from variants) derived from the variants generated by WGS and RNA-Seq data from all the samples in a cohort is used. However, it looks like only a global FDR is used in quality control. More stringent quality control for variant peptide identification should be used to remove potential false positives (PMID: 25357241, PMID: 32273506, PMID: 30610011) and how this affects SMAP performance should be evaluated.
2. The performance of SMAP is not well-demonstrated. First, only one large-scale TMT dataset is used. More datasets from different types of data (for example, label-free, iTRAQ datasets which have different complexity compared with the TMT-11plex data used in the current study) should be used to demonstrate the applicability of SMAP. Second, SMAP identified 66 mislabeled samples and 34 of them (~51.5%) had consistent correction results with using ATAC-Seq and Ribo-Seq data. However, the results for the remaining 32 samples (~48.5%) from SMAP are true or not is not clear. In addition, is the correction using SMAP accepted by the BrainGVEX working group to correct the data at <https://psychencode.synapse.org/Explore/Studies/DetailsPage?studyName=BrainGVEX>? Third, how SMAP performs compared with existing tools like COSMO is missing.
3. The common reference sample in each TMT experiment is not used to correct batch effect in peptide/protein quantification using BrainGVEX TMT-11plex data in this study. The strategy to use a common reference sample to remove batch effect is widely used in proteogenomic studies (PMID: 27251275, PMID: 27372738, PMID: 31031003, PMID: 31675502, PMID: 32059776, PMID: 32649874). How do the authors handle batch effect and how does batch effect affect the performance of SMAP?
4. Most of the variant peptides identified are only present in a few samples. How does SMAP handle missing value?
5. What is the minimum requirement of the number of identified variant peptides in a sample to use SMAP to get confident mislabeled sample identification?

Minor issues:

1. The authors claim that all raw mass spectrometry files, peptide and protein identification results, and genotypic data are available at the Synapse website (<https://psychencode.synapse.org/Explore/Studies/DetailsPage?studyName=BrainGVEX>), but I don't find both the raw MS data and identification results at the website.

2. Is there a published paper about how the proteomics data is generated? If not, it would be helpful to add method description for the data generation.
3. The m/z range on X-axis for TMT reporter ions is not correct in Figure 1C.
4. The original JUMPg pipeline consists of a multi-stage database search. Is the same search strategy used in the current study?
5. A supplementary table contains the identified variant peptides, their scores, spectra IDs, TMT experiment IDs, and reporter ion intensities should be provided.
6. What is the gene annotation used in variant annotation using ANNOVAR? RefSeq, Ensembl, or GENCODE?
7. In customized database construction, why do the authors use UniProt reviewed proteins as reference proteins rather than the matched reference proteins from the gene annotation used in variant annotation? The latter is commonly used in variant peptide identification. The authors should remove variant peptides which could be exactly mapped to any known protein isoforms in human.
8. Is there an automatic way to select an optimal value for parameter noise_level in SMAP?

We would like to thank the three reviewers for their thoughtful comments and efforts towards improving our manuscript. The point-by-point response to the comments is as follows:

Reviewer #1 (Expertise: proteogenomics, translational research): In this manuscript, the authors present an analysis pipeline for determining if there are sample mix-ups during proteogenomics analysis. They demonstrate that through correlation of variant-containing peptides, one can correct for sample mix-ups with high accuracy. The paper was clearly written, easy to follow, and the codebase and documentation seemed to be well put together.

Response: We thank the reviewer for positive comments about our manuscript and insightful views.

Comment #1: *First, unfortunately, there has been a recent publication in Patterns on a similar idea (PMID: 34036290). In that paper, the authors demonstrate the idea of sample-MS dataset matching with a broader scope, including several consortium datasets and participants. At the least, now, there would need to be a comparison of the SMAP approach against their COSMOS approach.*

Response: We appreciate the reviewer's constructive suggestions. As suggested by the reviewer, we performed a comparison of SMAP with COSMO using our matched transcriptomic and proteomic data (**Figure 1a**). A key step in COSMO is feature selection by choosing highly corrected gene and protein pairs. We found that a correlation coefficient of 0.5 is used as a default parameter in COSMO. When applying this setting to our omic data, no features were selected partly due to about 20% mis-labeled samples in our proteomic data base on the corrections by SMAP. To obtain an optimal number of features, we examined correlation coefficients ranging from 0.2 to 0.55, and found that the correlation coefficient of 0.35 produced the best performance when considering the number of selected features and matched samples. (**Figure 1b**).

By using the selected correlation coefficient of 0.35, COSMO detected 68 mis-labeled samples out of 288 samples in our proteomic data. We found that SMAP and COSMO made the same correction for 270 (94%) samples, including 218 samples matched to its original ID, and 52 mis-labeled samples but corrected to the same identity by both methods. In the remaining 18 (6%) samples that showed inconsistent correction between COSMO and SMAP (**Figure 1c**), we observed that 4 samples matched to the original identity were "mis-assigned" to the other identity by COSMO, whereas SMAP made no such correction. For example, SMAP showed no correction for a sample, *S2015_1504*, with a high Cscore and Δ Cscore (**Figure 1d**); however, it was assigned to a new sample identity by COSMO. SMAP "mis-assigned" 2 samples, but COSMO showed the samples had high correlations. For example, the sample, *S2015_1477*, showed a high significant correlation of the expression levels between protein and RNA with the original sample ($r = 0.65$, $p = 1.9 \times 10^{-14}$) (**Figure 1e**). SMAP and COSMO adjusted two samples into different samples with high Cscore and correlation coefficient, respectively. Two samples adjusted into different sample identities with high scores. In addition, we also found that 10 samples that cannot be corrected by both SMAP and COSMO because of low Cscore of 1.30 ± 0.12 (2.83 ± 0.55 , on average, from 218 matched samples) and low correction coefficient of 0.10 ± 0.04 (0.45 ± 0.09 , on average, from 218 matched samples), presumably due to no matched samples between proteomic and transcriptomic data.

Accordingly, we added a new result section to the revised manuscript:

“Performance evaluation of SMAP compared to COSMO

*To assess the performance of SMAP, we also analyzed our data using COSMO, a program recently developed to correct sample mislabeling in omic data through correlating mRNA and protein expression levels. A key step in the COSMO is feature selection by selecting highly corrected gene and protein pairs. We found that a correlation coefficient of 0.5 is used as a default parameter in COSMO. When applying this setting to our omic data, no features were selected in our data partly due to about 20% were mis-labeled samples in our proteomic data. To obtain an optimal number of features, we examined correlation coefficients ranging from 0.2 to 0.55, and found that the correlation coefficient of 0.35 produced the best performance when considering the number of selected features and matched samples (**Supplementary Figure 6a, b**).*

*By using the selected correlation coefficient of 0.35, COSMO detected 68 mis-labeled samples out of a total of 288 samples in our proteomic data. We found that SMAP and COSMO made the same correction for 270 (94%) samples, including 218 samples matched to its original ID, and 52 mislabeled samples but corrected to the same identity by both methods. In the remaining 18 (6%) samples that shows inconsistent correction between COSMO and SMAP (**Supplementary Figure 6c**), we observed that 4 samples matched to the original identity by SMAP were “mis-assigned” to other identities by COSMO. For example, SMAP made no change for the sample S2015_1504, with a high Cscore and Δ Cscore (**Supplementary Figure 6d**); however, it was re-assigned to a new sample identity by COSMO. Conversely, SMAP also “mis-assigned” 2 samples, but COSMO indicated that the samples had high correlations. For example, the sample S2015_1477, showed a correlation of the expression levels between protein and RNA with the original sample ($r = 0.65$, $p = 1.9 \times 10^{-14}$) (**Supplementary Figure 6e**). Two samples were adjusted into different sample identities by SMAP and COSMO with high scores. In addition, we found that 10 samples cannot be corrected by both SMAP and COSMO because of a low average Cscore of 1.30 ± 0.12 (2.83 ± 0.55 , on average, from 218 matched samples) and a low average correction coefficient of 0.10 ± 0.04 (0.45 ± 0.09 , on average, from 218 matched samples), presumably due to no matched samples between proteomic and transcriptomic data.”*

Comment #2: *Second, and unrelated to the first comment, I feel that the scope of the work is quite focused on one problem (sample mix-ups in proteomics) and would need to be significantly broadened for this journal.*

Response: We agree with the reviewer that the scope of this work is primarily focused on proteomic data. The reviewer is also right that mislabeling issue is prevalent in almost every omic data. Although SMAP was designed for the proteomic data, the concept of leveraging genotypic data for sample correction can be applied to other omic data as well. The genotypic data of each sample can be readily extracted from all massive sequencing data, ranging from whole-genome DNA sequencing, epigenome (e.g., ATAC-seq), to transcriptome (bulk RNA and single-cell RNAseq). However, extracting genotypes from MS data presents a much greater challenge than

that from the sequencing data as we need to convert MS signals into amino acids, which makes the paper truly unique.

As suggested by the reviewer, we added the following text to the discussion in the revised manuscript:

“Correction of sample mislabeling is paramount to all large-scale omic studies. The SMAP program centers on sample correction for proteomic data by leveraging genotypic data. This concept of using genotypic data for sample correction can be applied to other omic data. The genotypes of each sample can be readily extracted from all massive sequencing data, ranging from whole-genome DNA sequencing, epigenome (e.g., ATAC-seq), to transcriptome (bulk RNA and single-cell RNAseq). However, inferring genotypic data from MS data is a much greater challenge than that from sequencing data as we need to convert the MS signals into amino acids and then convert amino acids into genotypes. More importantly, we proposed a novel scoring scheme in which a combined score is generated from two scores: (i) Cscore, which is a measure of the goodness of fit of genotypes inferred from proteomic data to reference genotypes; (ii) Δ Cscore: is a measure of the specificity of the fit. This new score scheme can be adapted to any other algorithms for sample correction.”

Reviewer #2 (Expertise: Genomics, bioinformatics): The authors present a method to identify samples with mismatched labels in multi-omics studies, where both proteomics and genomic/genetic datasets are available. Specifically, the authors build upon their previous work (the JUMPG framework, which in turn builds on the JUMP database search algorithm). The method importantly fills the gap in current methods that address isobaric labeling datasets, such as TMT and iTRAQ. It addresses an important concern (sample mismatches) which has been receiving necessary attention recently, by extending the remediation strategies to a new data type.

Response: We thank the reviewer for the expert advice. We are also encouraged that the reviewer found our study addresses an important concern in a large-scale proteomics study.

Comment #1: *While the central concepts and strategies can be understood, the authors need to explicitly define many essential terms, clean up several typographical errors and add more detail to the methods.*

Response: We would like to thank the reviewer for the careful reading. We have thoroughly revised our manuscript to eliminate any potential typos and clarify all terms used in the manuscript. We believe that the quality of the revised manuscript has been significantly improved.

Please address the following concerns:

Comment #2:-*Lines 106-112: Allelic information – Beyond the quantile argument, can't you also determine the intensities of the non-mutant peptides that overlap that region, and compare intensities to determine AA, AB, or BB? So, if the mutant and non-mutant peptides occur in nearly equal intensities in the same sample, you have a heterozygous mutation.*

Response: We appreciate the reviewer for raising this constructive comment. The reviewer is correct that we can detect both mutant peptides and their corresponding non-mutant peptides that overlap that region from the 10 samples in a TMT batch (**Figure 2a**). Although non-mutant and/or mutant peptides can be detected by mass spectrometry (MS), they typically exhibit different magnitudes of the absolute intensity due to the amino acid composition of the peptide, which influence intensity of the fragment ions in the MS/MS spectra. For example, the mutant peptide has about double the signal compared to a non-mutant peptide (6×10^5 vs 3×10^5), (**Figure 2b**) (Dongré et al., 1996, *Journal of the American Chemical Society*). Thus, we did not use the nearly equal absolute intensities of mutant and non-mutant peptides to infer whether there is a heterozygous mutation in the sample. However, non-mutant and mutant peptides have comparable relative intensities (~ 0.5) after standardization (**Figure 2c**). We used 2nd quartile (Q2: 0.25-0.5) and 3rd quartile (Q3: 0.5-0.75) of the standardized data to infer the heterozygous genotype.

In addition, we only used the intensity of mutant peptides in this study to infer the genotype because non-mutant peptides could potentially produce a biased measure due to the presence of shared peptides produced by gene families in the genome (**Figure 2d**). The abundance of a non-mutant peptide could be contributed by multiple homologous proteins, whereas the abundance of a mutant peptide tends to be only contributed by the unique mutant peptide itself because it is less likely to have the same mutation in both paralogous proteins. To approve this,

we performed a correlation analysis of the abundance between mutant and non-mutant peptides. We expect a strong negative correlation between mutant and non-mutant peptides. We found a negative correlation ($r = -0.92$; $p = 1.8 \times 10^{-4}$) between a mutant peptide (FYLLGGPTSIR) and non-mutant peptide (FYLLGGPVSVR), (**Figure 2e**). However, we found ~20% of the mutant and non-mutant peptide pairs showed a positive correlation (**Figure 2f**). For example, a non-mutant peptide (EFHPVLK) showed a strong correlation ($r = 0.87$; $p = 1.2 \times 10^{-3}$) with its corresponding mutant peptide (ELHPVLK), (**Figure 2g**). By manual examination, we found that the non-mutant peptide is not accurately quantified as we expected.

Comment #3:-Line 115: Please define the concordance score: it is not the percentage of overlapping genotypes. So how is it defined?

Response: The concordance score (i.e., Cscore) is defined as the percentage of matched genotypes. The formula is shown as follows,

$$Cscore = \frac{\text{Number of matched genotypes}}{\text{Total number of detected genotypes}}$$

We added the definition in the result section 2.1.3 of the revised manuscript.

Comment #4: -Lines 129-130: How exactly are the authors are shuffling genotypes: are they starting with genotype dosages for the alternate allele, and then randomly mutating to one of the other two values? For example, if for the true sample, SNP i has an alternate allele dosage of 2, are they mutating it randomly to 0 or 1? Or just randomly choosing SNP dosages from the other samples? In the latter case, one may randomly choose the same SNP dosage as before (in the above example, SNP i dosage = 2 may be replaced with SNP i dosage = 2 from the other samples). This would result in an overestimation of the genotype substitution rate. Or are they doing something else? Figure 2 is confusing as it implies that the bases themselves are randomly bent mutated (in both haplotypes?). Please clarify this, as the exact definition has implications for how the shuffling process would reduce concordance.

Response: We agree with the reviewer that our randomly choosing a SNP would result in an overestimation of the genotype substitution rate. As suggested by the reviewer, we modified our shuffling strategy by excluding the same SNP dosage (see Step 4 below). To clarify our shuffling strategy, we present the strategy in a new diagram (**Figure 3**):

Step 1: Calculate the frequency of each allele;

Step 2: Randomly select a sample i ;

Step 3: Randomly select a SNP j ;

Step 4: Select another allele with a probability of allele frequency (calculated in Step 1);

Step 5: Swap the SNP j in the sample i with selected allele in step 4;

Step 6: Repeat steps 1-5 to generate a simulated data with a certain percentage of error rate;

With this strategy, we re-evaluated the results and updated the results in the previous **Figure 4** below (**Figure 2** in the revised manuscript).

Accordingly, we added the follow text to the result section 2.2.

“We generated a simulated data by six steps (**Supplementary Figure 4**): (i) calculating the frequency of each allele; (ii) randomly selecting a sample i ; (iii) randomly select a SNP j ; (iv) selecting another allele with a probability of allele frequency; (v) swapping the SNP j in the sample i with selected allele in the step 4; and (vi) repeating steps 1-5 to generate a certain percentage of error rate.”

Comment #5:-Lines 154-171: The threshold and S/N ratio cutoffs are defined in terms of homozygous alleles only. In the EM method, the authors identify only two peaks. Will this choice of cutoffs produce a bias against the detection of heterozygotes? That will then have an impact on the allele frequencies of the SNPs that can be detected. What is the MAF distribution of detected SNPs?

Response: The reviewer is right that the S/N is calculated based on the intensities of two homozygous alleles, which may produce a bias to the detection of heterozygotes. However, we attempted to avoid this issue with two solutions. The first solution is that we used the quartile method to assign the genotype dosage, which allows a tolerance for mis-assignment. In the case of the detection of heterozygotes, 2nd quartile (Q2: 0.25-0.5) and 3rd quartile (Q3: 0.5-0.75) of the standardized data were used. The second solution is that we also used the dosage information in the genotype file as prior knowledge to assign the inferred genotypes (**Figure 5**). For example, if there are only two genotypes (e.g., mutant and non-mutant alleles) in the genotype file, the program will use a cutoff of 0.5 to assign the two genotypes by excluding the possibility of inferring a heterozygous genotype.

Figure 5. Strategy of genotype assignment.

For the EM method, we filtered false mutant peptides with the minimum intensity being significantly greater than that of other mutant peptides (2 SD; $p < 1\%$). The EM method produced two distributions: (1) a distribution of true noise signal; (2) a distribution of false noise signal. If the minimum intensity falls in the second distribution, it is highly likely to be a false identification of mutant peptides. As showed in the manuscript, only a small proportion (24/322) of mutant peptides were filtered.

We used the MAF of 0.05 as a filtering, meaning that those SNPs with MAF > 0.05 were used for sample correction, but it is optional.

We add the following text to the method section to clarify the genotype assignment:

*“We also used the dosage information in the genotype file as prior knowledge to infer genotypes (**Supplementary Figure 3**). For example, if there are only two genotypes (e.g., mutant and non-mutant alleles) in the genotype file, the program will use a cutoff of 0.5 to assign the two homozygous alleles by excluding the possibility of inferring a heterozygous genotype.”*

Comment #6:-Lines 265-283: *What are the error rates from the previous pipelines (JUMP, JUMPG)? While a comprehensive comparison is not necessary, could you comment on the rate of occurrence of ambiguous variant peptides in your datasets? How was the filtering carried out? Stating this in the manuscript would be very helpful.*

Response: As suggested by the reviewer, we clarified the error rate in the revised manuscript. In this study, we initially used the FDR rate < 5% for variant peptide identification with the target-decoy strategy. In JUMPG, a two-stage FDR method is used: in the first stage, the MS/MS data are searched against a reference protein database, and the confidently identified spectra are removed; in the second stage, the remaining spectra are searched against the variant protein database, and the FDR for variant peptides is calculated based on the second stage search results.

To filter variant peptides, PSMs are first filtered by user-specified parameters (e.g., minimum peptide length and minimum search score), then by precursor ion mass accuracy. The resulting PSMs are further grouped by precursor ion charge state and tryptic ends and then filtered by matching scores (Jscore and Δ Jscore) to achieve the user-specified level of FDR. We used a peptide FDR < 5% in this study.

We add the following text to the method section in the revised manuscript:

“In JUMPG, a two-stage FDR method is used: in the first stage, the MS/MS data are searched against a reference protein database, and the confidently identified spectra are removed; in the second stage, the remaining spectra are searched against the variant protein database, and the FDR for variant peptides is calculated based on the second stage search results.”

“To filter variant peptides, PSMs are first filtered by user-specified parameters (e.g., minimum peptide length and minimum search score), then by precursor ion mass accuracy. The resulting PSMs are further grouped by precursor ion charge state and tryptic ends and then filtered by matching scores (Jscore and Δ Jscore) to achieve a peptide FDR < 5%.”

Comment #7:-Supplementary Figure 3: *What are the unmatched samples? Are those internal controls? Have all the samples in the first batch been matched?*

Response: For each sample, we performed a comparison against all samples in the genotype data (**Figure 6a**). For example, when a sample i is selected for comparison, it is compared to all n samples in the genotype data, which produces one “matched” sample with good Cscore and Δ Cscore, and remaining $n-1$ “unmatched” samples. In the case of the internal standard, it has one “matched” sample even though the assigned identity is incorrect. To clarify, only one sample (red dot) in the first batch has been corrected to a new sample identity highlighted in red

dot in **Figure 6b**.

Minor edits:

Comment #8:-Figure 4a: Label for the orange swatch (4th from the top) is incorrect – should be “MS-specific”.

Response: We thank the reviewer for careful reading. We have corrected the label in the revised manuscript.

Comment #9:-Figure 2 caption: Lines 350-351: Maybe change the names of the samples: “selected sample with highest score” and “select sample” are too similar. Make the latter “true sample”.

Response: We have corrected the word as the reviewer suggested.

Comment #10:-Figure 2a: What does the H represent in the figure? Heterozygote?

Response: The reviewer is right that the H represents heterozygote. We clarified it in the figure legend.

Comment #11:-Lines 32-33: The wording seems to imply that 18.8% of the mismatched samples were found and corrected (and that 71.2% of mismatched samples were not). The

actual situation is that 18.8% of all samples were found to be mismatched, and that these were detected using SMAP. Please clarify the language here.

Response: We identified 22.9% (66/288) are mismatched, 18.8% (54/288) were corrected.

Comment #12:-Line 91: FASTQ file, not FASTAQ file

Response: We corrected “FASTQ” in Line 91.

Comment#13:-Figure 1c: Why is sample 11 shown in red? Are the authors highlighting anything in particular? Maybe Sample 3 should be red instead?

Response: We updated “Figure 1C” as suggested by the reviewer. We initially attempted to indicate that sample 11 is the internal standard, but the reviewer’s suggestion makes more sense.

Comment #14:-Line 136: The 31 internal controls are “mixture(s) of the 288 samples”? How are they “mixed”? Please explain.

Response: We mixed all proteins of 288 samples, and used the mixed sample as the internal standard in each batch.

Comment #15:-Line 142: Mixed samples, or mismatched samples? I think mismatched is intended, so please use consistent terminology.

Response: We corrected it with “mis-labeled samples”.

Comment #16:-Line 149: This reference to Figure 3a should be probably be removed. The numbers just quoted (~20,000 SNPs) do not appear in Figure 3a.

Response: We corrected it in the revised manuscript.

Comment #17:-Figure 3b: The black dotted line should end at the red threshold, right? The ratio is of the BB to AA intensities (with the AA intensity being defined by the red line)?

Response: The figure has been updated.

Reviewer #3 (Expertise: proteogenomics):

In this manuscript, Li et al. present a novel pipeline named SMAP that makes use of both genomics data and MS-based proteomics data to identify and correct mislabeled samples in proteogenomic studies. Using multi-omics data in proteogenomics studies for mislabeled sample identification and correction is not new. A published tool called COSMO from the precisionFDA NCI-CPTAC Multi-omics Enabled Sample Mislabeled Correction Challenge has shown the value to combine proteomics data with other omics data including genomics data in mislabeled sample identification and correction (PMID: 34036290). However, the use of variant peptide identification from proteomics in SMAP is somewhat novel. The authors tested SMAP on a large-scale proteogenomic dataset in which the proteomics data contains 29 batches of TMT11-plex data and compared SMAP with the results using other omics data from the same cohort. This pipeline may be useful in proteogenomics studies. However, there are a few major and minor issues needed to be addressed.

Response: We appreciate this reviewer for all critical comments. The COSMO program is the first, to the best of our best knowledge, a tool for correcting the sample mixed-ups. Although the COSMO uses expression levels of protein and mRNA, which is different from SMAP using genotypes, it is very a good benchmark for comparison of the performance of SMAP.

Major issues:

Comment #1: *Variant peptide identification in proteogenomics has been known to be error-prone especially in searching large non-sample specific customized databases in which most sequences in the search space are false. In this study, a single customized database (17,844,001 theoretical peptides from variants) derived from the variants generated by WGS and RNA-Seq data from all the samples in a cohort is used. However, it looks like only a global FDR is used in quality control. More stringent quality control for variant peptide identification should be used to remove potential false positives (PMID: 25357241, PMID: 32273506, PMID: 30610011) and how this affects SMAP performance should be evaluated.*

Response: We thank the reviewer for the expert comment. The reviewer is right that we used JUMPg for variant peptide identification and a two-stage FDR method was used for variant peptide identification. In the first stage, the MS/MS data were searched against a reference protein database, and the confidently identified spectra were removed. In the second stage, the remaining spectra were searched against the protein database, and the FDR for variant peptides was calculated based on the second stage search results.

In addition, we agree with the reviewer that a global FDR will underestimate the FDR and is prone to false-positive variant peptide identifications. We also evaluated the separate FDR method (Karpova, M. A. et al., 2014, *J. Proteome Res.* Li, J. et al., 2011, *Mol. Cell Proteom.*) by calculating the global FDR, which uses a combination of known and variant peptides, and the local FDR, which uses variant peptides only. We detected 482 and 442 variant peptides per batch, on average, at the global FDR < 1% and the local FDR < 1%, respectively. The global FDR slightly inflate the number of variants.

In addition to the initial FDR we also filtered peptides using the signal-to-noise ratio (S/N) and minimum intensity, which would increase the confidence of variant peptides. SMAP is capable of

tolerating up to ~80% of genotypic mis-assignment, as shown in our simulation study. Owing to the above two reasons, we used a relatively loose FDR cutoff (< 5% peptide FDR) in the initial step of the peptide identification.

Comment #2: *The performance of SMAP is not well-demonstrated. First, only one large-scale TMT dataset is used. More datasets from different types of data (for example, label-free, iTRAQ datasets which have different complexity compared with the TMT-11plex data used in the current study) should be used to demonstrate the applicability of SMAP. Second, SMAP identified 66 mislabeled samples and 34 of them (~51.5%) had consistent correction results with using ATAC-Seq and Ribo-Seq data. However, the results for the remaining 32 samples (~48.5%) from SMAP are true or not is not clear. In addition, is the correction using SMAP accepted by the BrainGVEX working group to correct the data at <https://psychencode.synapse.org/Explore/Studies/DetailsPage?studyName=BrainGVEX?> Third, how SMAP performs compared with existing tools like COSMO is missing.*

Response: We appreciate the reviewer's constructive suggestions. As suggested by the reviewer, we performed a comparison of SMAP with COSMO using our matched transcriptomic and proteomic data (**Figure 1a**). A key step in COSMO is feature selection by choosing highly corrected gene and protein pairs. We found that a correlation coefficient of 0.5 is used as a default parameter in COSMO. When applying this setting to our omic data, no features were selected partly due to more than 20% mis-labeled samples in our proteomic data base on the corrections by SMAP. To obtain an optimal number of features, we examined correlation coefficients ranging from 0.2 to 0.55, and found that the correlation coefficient of 0.35 produced the best performance when considering the number of selected features and matched samples. (**Figure 1b**).

By using the selected correlation coefficient of 0.35, COSMO detected 68 mis-labeled samples out of 288 samples in our proteomic data. We found that SMAP and COSMO made the same correction for 270 (94%) samples, including 218 samples matched to its original ID, and 52 mis-labeled samples but corrected to the same identity by both methods. In the remaining 18 (6%) samples that showed inconsistent correction between COSMO and SMAP (**Figure 1c**), we observed that 4 samples matched to the original identity were "mis-assigned" to the other identity by COSMO, whereas SMAP made no such correction. For example, SMAP showed no correction for a sample, *S2015_1504*, with a high Cscore and Δ Cscore (**Figure 1d**); however, it was assigned to a new sample identity by COSMO. SMAP "mis-assigned" 2 samples, but COSMO showed the samples had high correlations. For example, the sample, *S2015_1477*, showed a high significant correlation of the expression levels between protein and RNA with the original sample ($r = 0.65$, $p = 1.9 \times 10^{-14}$) (**Figure 1e**). SMAP and COSMO adjusted two samples into different samples with high Cscore and correlation coefficient, respectively. Two samples adjusted into different sample identities with high scores. In addition, we also found that 10 samples that cannot be corrected by both SMAP and COSMO because of low Cscore of 1.30 ± 0.12 (2.83 ± 0.55 , on average, from 218 matched samples) and low correction coefficient of 0.10 ± 0.04 (0.45 ± 0.09 , on average, from 218 matched samples), presumably due to no matched samples between proteomic and transcriptomic data.

Accordingly, we added a new result section to the revised manuscript:

“Performance evaluation of SMAP compared to COSMO

*To assess the performance of SMAP, we also analyzed our data using COSMO, a program recently developed to correct sample mislabeling in omic data through correlating mRNA and protein expression levels. A key step in the COSMO is feature selection by selecting highly corrected gene and protein pairs. We found that a correlation coefficient of 0.5 is used as a default parameter in COSMO. When applying this setting to our omic data, no features were selected in our data partly due to about 20% were mis-labeled samples in our proteomic data. To obtain an optimal number of features, we examined correlation coefficients ranging from 0.2 to 0.55, and found that the correlation coefficient of 0.35 produced the best performance when considering the number of selected features and matched samples (**Supplementary Figure 6b**).*

*By using the selected correlation coefficient of 0.35, COSMO detected 68 mis-labeled samples out of a total of 288 samples in our proteomic data. We found that SMAP and COSMO made the same correction for 270 (94%) samples, including 218 samples matched to its original ID, and 52 mislabeled samples but corrected to the same identity by both methods. In the remaining 18 (6%) samples that shows inconsistent correction between COSMO and SMAP (**Supplementary Figure 6c**), we observed that 4 samples matched to the original identity by SMAP were “mis-assigned” to other identities by COSMO. For example, SMAP made no change for the sample S2015_1504, with a high Cscore and Δ Cscore (**Supplementary Figure 6d**); however, it was re-assigned to a new sample identity by COSMO. Conversely, SMAP also “mis-assigned” 2 samples, but COSMO indicated that the samples had high correlations. For example, the sample S2015_1477, showed a correlation of the expression levels between protein and RNA with the original sample ($r = 0.65$, $p = 1.9 \times 10^{-14}$) (**Supplementary Figure 6e**). Two samples were adjusted into different sample identities by SMAP and COSMO with high scores. In addition, we found that 10 samples cannot be corrected by both SMAP and COSMO because of on average low Cscore of 1.30 ± 0.12 (2.83 ± 0.55 , on average, from 218 matched samples) and low correction coefficient of 0.10 ± 0.04 (0.45 ± 0.09 , on average, from 218 matched samples), presumably due to no matched samples between proteomic and transcriptomic data.”*

For those 32 samples (~48.5%) that did not match both Ribo-seq and ATAC-seq, we summarized the results in **Figure 8**, which include 9 samples with the same calibration between proteomics and ATAC-seq, 5 between proteomics and Ribo-seq, and 6 proteomic-specific calibrations (**Fig. 4a in the manuscript**). In addition, SMAP identified 12 samples that could be mislabeled but cannot provide a confident calibration because of low score.

As suggested by the reviewer, we also attempted to perform sample correction by SMAP using an additional data set. We downloaded a mouse liver proteomic data set with the accession number of PXD002801 in the PRIDE database, which was also used in the COSMO paper. However, we were only able to obtain deep whole-genome sequencing data of 8 parental strains and shallow genotype data for all Diversity Outbred (DO) and Collaborative Cross (CC) mice. With the deep SNP data, our SMAP identified 364 variant peptides, on average, across the entire population. However, the correction cannot be made due to the lack of deep SNP data from each strain. We believe that SMAP is able to correct the mis-labeled samples in batches #14 and #15 as indicated by COSMO once the deep SNP sequencing data of DO and CC strains are available.

Comment #3: *The common reference sample in each TMT experiment is not used to correct batch effect in peptide/protein quantification using BrainGVEX TMT-11plex data in this study. The strategy to use a common reference sample to remove batch effect is widely used in proteogenomic studies (PMID: 27251275, PMID: 27372738, PMID: 31031003, PMID: 31675502, PMID: 32059776, PMID: 32649874). How do the authors handle batch effect and how does batch effect affect the performance of SMAP?*

Response: SMAP performs batch-wise sample correction. Therefore, we believe the batch effect might have a trivial impact on its performance. Nonetheless, we examined the impact of the batch effect on SMAP performance. We found that our BrainGVEX TMT-11plex data showed a strong batch effect even after correcting by the common reference sample (**Figure**

9a). To further remove the batch effect, we used the PEER (probabilistic estimation of expression residuals), a software program to remove all potential hidden factors (**Figure 9b**). We found that SMAP showed a high consistency of Cscore before and after batch effect removal ($r = 0.98$, $p = 2.2 \times 10^{-16}$), (**Figure 9c**).

Comment #4: Most of the variant peptides identified are only present in a few samples. How does SMAP handle missing value?

Response: There are two types of missing values in a typical TMT data set: (1) a small proportion of missing values from intra-batch samples due to the limitation of MS detection; and (2) a large proportion of missing values from inter-batch samples when combined. For missing values within a batch, SMAP assigns a noise value for all missing values, which is the median of the minimum intensity of all variant peptides (**See Comment #12**). Since it is a noise value, a non-mutant allele is assigned to the sample. Since SMAP performs batch-wise sample correction, the missing values from inter-batch samples have trivial impact on performance of SMAP.

Comment #5: What is the minimum requirement of the number of identified variant peptides in a sample to use SMAP to get confident mislabeled sample identification?

Response: To evaluate the minimum number of identified variant peptides, we first curated a data set including those 10 samples that all match the original ID. We then examined whether SMAP was able to assign the original identity by reducing the number of variant peptides from 100 to 10, successively, by numbers of 10. We found that a minimum of 40 variant peptides is required to achieve >90% successful correction rate.

Minor issues:

Comment #6: The authors claim that all raw mass spectrometry files, peptide and protein identification results, and genotypic data are available at the Synapse website

(<https://psychencode.synapse.org/Explore/Studies/DetailsPage?studyName=BrainGVEX>), *but I don't find both the raw MS data and identification results at the website.*

Response: The raw MS data described in this manuscript are now available via the PsychENCODE Knowledge Portal with an accession number of syn26231732. Considering that the data only can be access by a registered user, we provided the editor with the information on how to access the data for un-registered user.

Comment #7: *Is there a published paper about how the proteomics data is generated? If not, it would be helpful to add method description for the data generation.*

Response: We provided method description about data generation in the section of "METHODS: TMT QUANTITATION" on PsychENCODE Knowledge Portal website (<https://psychencode.synapse.org/Explore/Studies/DetailsPage?study=syn4590909>).

Comment #8: *The m/z range on X-axis for TMT reporter ions is not correct in Figure 1C.*

Response: We thank the reviewer for careful reading. We have corrected it in the revised manuscript.

Comment #9: *The original JUMPg pipeline consists of a multi-stage database search. Is the same search strategy used in the current study?*

Response: The reviewer is right that we used the same multi-stage search strategy.

Comment #10: *A supplementary table contains the identified variant peptides, their scores, spectra IDs, TMT experiment IDs, and reporter ion intensities should be provided.*

Response: As suggested by the reviewer, a new **Supplementary Table 2** has been added to the revised manuscript, including SNP chromosome, SNP position, reference allele, alternative allele, peptide, scan number (batch#, fraction#, and scan#), JUMP matching score (i.e., Jscore and dJn), and reporter ion intensities.

Comment #11: *What is the gene annotation used in variant annotation using ANNOVAR? RefSeq, Ensembl, or GENCODE?*

Response: We used the UCSC gene annotation for ANNOVAR. We clarified it in the revised manuscript.

Comment #12: *In customized database construction, why do the authors use UniProt reviewed proteins as reference proteins rather than the matched reference proteins from the gene annotation used in variant annotation? The latter is commonly used in variant peptide*

identification. The authors should remove variant peptides which could be exactly mapped to any known protein isoforms in human.

Response: We appreciate the reviewer for the suggestion. To remove all potential known protein isoforms, we downloaded two comprehensive protein databases: (1) 181,1774 un-reviewed proteins from the TrEMBL database; (2) 63,691 proteins from the UCSC database. By comparing our previous results reviewed protein database to all variant peptides identified with two larger databases (i.e., TrEMBL and UCSC), we removed 58 unique variant peptides from **Supplementary Table 1 (Figure 10)**.

Comment #13: Is there an automatic way to select an optimal value for parameter `noise_level` in SMAP?

Response: As suggested by the reviewer, we implemented a new function in the code to assign an optimal noise level in the SMAP program by automatically calculating the median value of the minimum intensity in all identified variant peptides. This will be used as default parameter, but users can set a specified value.

Reviewers' Comments:

Reviewer #2:

Remarks to the Author:

In reviewing the updates to the manuscript, I appreciate the significant additional work carried out by the authors in response to the reviewer comments. However, I still feel that some points should be cleared up:

1. Shuffling process, line 140: I appreciate the update to the methods. I still think there is a little ambiguity that should be cleared up:
 - a. Firstly, when you say allele frequency, do you mean genotype frequency (i.e. frequency of AA, AB and BB)? Or are you changing each allele separately with a certain error rate?
 - b. When you select another allele according to the sample allele frequency, are you choosing from different alleles, or possibly also the same allele? Take an example. If the correct genotype is 2 (=BB), and the genotype frequencies are AA = 0.2, AB = 0.3, BB = 0.5, are you selecting AA and AB only or any of the three? If you are only selecting AA and AB are you renormalizing (AA = 0.2/(0.2+0.3) for eg.) or not? The choices depend on what error scenario you are simulating. If you are simulating simple genotyping error you should include all three with an error that is independent of the genotypes selected. If you are simulating only sample mix-up errors, then you should probably select all three genotypes according to their allele frequencies (which is what I think you did). But it might be important to state that the true error rate may be overestimated as you might replace AA=2 with AA=2 in the above example. In any case, please be clearer on this point.
2. Figure 3d: is the x-axis the expression intensity noise threshold or is it the expression intensity of the SNP? It makes more sense if it is the expression intensity of the SNP. If it is the noise threshold, then increasing the noise threshold should decrease the number of SNPs observed, correct (because less variants are called as true SNPs)? Please clarify this in the text and caption.
3. Supplementary figure 2 and 4 captions are mixed up
4. Supplementary figure ??: Strategy of genotype assignment  Rows are shown with index j and are called samples. Are these SNP samples? Maybe they should just be called SNPs? The columns are "Genotypes", are they the samples? But then you select out SNP j and call it Genotype i? This figure should be labeled more clearly. And maybe replace "Genotype number" with "Number of unique genotypes in sample database". That makes things more transparent.
5. Supplementary Figure 6b: How can the number of features be 0, yet the number of matched samples be > 0%?
6. Line 282: "Qualitative alleles from genomic and transcriptomic data" This is not entirely true. Genomic and transcriptomic allele information comes from stacking reads at a variant locus. For example, in the case of a heterozygous allele, enough reads of the variant need to be stacked up in order for the pipeline (say, GATK) to properly call the variant. So it is a quantitative process.

Minor edits:

Throughout text: "sample mixed-up"  "sample mix-up"

Line 37: "...chromatin sequencing data"  "chromatin sequencing (ATAC-seq) data"

Formula between lines 106 and 107:

$$y_i = (x_i - \min_{i'}(x_{i'})) / (\max_{i'}(x_{i'}) - \min_{i'}(x_{i'}))$$

Showing that the min and max functions are carried out with respect to the samples i clarifies that you are finding the max and min across samples, and that the scaling is bringing all samples to the same scale.

Line 137: "exacted"  "extracted"
Line 217: Unnecessary newline character

Prashant Emani

Reviewer #3:

Remarks to the Author:

All my comments have been well addressed. I don't have any further comments.

The point-by-point response to the reviewer's comments is as follows:

Reviewer #2 (Remarks to the Author):

In reviewing the updates to the manuscript, I appreciate the significant additional work carried out by the authors in response to the reviewer comments. However, I still feel that some points should be cleared up:

Response: We thank the reviewer for all critical comments and suggestions. We are also grateful for the reviewer's careful reading.

Comment #1: *Shuffling process, line 140: I appreciate the update to the methods. I still think there is a little ambiguity that should be cleared up:*

- a. *Firstly, when you say allele frequency, do you mean genotype frequency (i.e., frequency of AA, AB and BB)? Or are you changing each allele separately with a certain error rate?*

Response: The reviewer is right that we refer to genotype frequency. We made a change of "allele frequency" into "genotype frequency" in the revised manuscript (please see the clear copy on page 5, line 132). To clarify it, we have also made the necessary corrections across the entire manuscript. As the reviewer's suggestion, we have clarified it in the revised result section and the legend of Supplementary Figure 4 as follows,

"We generated a simulated dataset in six steps (**Supplementary Fig. 4**): (1) estimating the frequency of each genotype across all samples; (2) randomly selecting a sample i ; (3) randomly selecting a genotype j in the sample i ; (4) choosing another genotype with a frequency estimated in step 1; (5) swapping the genotype j in the sample i with a chosen genotype in step 4; and (6) repeating steps 1-5 to generate a simulated dataset with a certain percentage of the error rate (e.g., 10%, 20%, 40%, and 80%)."

- b. *When you select another allele according to the sample allele frequency, are you choosing from different alleles, or possibly also the same allele? Take an example. If the correct genotype is 2 (=BB), and the genotype frequencies are AA = 0.2, AB = 0.3, BB = 0.5, are you selecting AA and AB only or any of the three? If you are only selecting AA and AB are you renormalizing (AA = 0.2/(0.2+0.3) for eg.) or not? The choices depend on what error scenario you are simulating. If you are simulating simple genotyping error you should include all three with an error that is independent of the genotypes selected. If you are simulating only sample mix-up errors, then you should probably select all three genotypes according to their allele frequencies (which is what I think you did). But it might be important to state that the true error rate may be overestimated as you might replace AA=2 with AA=2 in the above example. In any case, please be clearer on this point.*

Response: We really appreciate the reviewer's thoughtful comment. Following the example provided by the reviewer, if the correct genotype is 2 (=BB), and the genotype frequencies are AA = 0.2, AB = 0.3, BB = 0.5, we use AA = 0.2 and AB = 0.5 **only** instead of renormalizing AA = $0.2 / (0.2+0.5) = 0.29$ and AB = $0.5 / (0.2+0.5) = 0.71$, which, we thought, is the reviewer's previous suggestion. Another reason to keep the original genotype frequency is that we estimate the genotype frequency from the entire genotypic data instead of each genotype across all samples. We hope we have addressed the reviewer's concern properly.

Comment #2: *Figure 3d: is the x-axis the expression intensity noise threshold or is it the expression intensity of the SNP? It makes more sense if it is the expression intensity of the SNP. If it is the noise threshold, then increasing the noise threshold should decrease the number of SNPs observed, correct (because less variants are called as true SNPs)? Please clarify this in the text and caption.*

Response: The reviewer raised a very good question. The x-axis represents a cutoff value of the minimum intensity of variant peptides. We made a figure below to explain why raising the cutoff value increases the number of SNPs and decreases Cscore in **Figure 3d**. For example, there are 10 variant peptides detected in the analysis. Each variant peptide contains different expression values across 10 samples (**panel a**). In the case where both reference and alternative genotypes are observed in 10 samples, the samples with a relatively high expression value could be derived from the variant peptide containing an alternative genotype, whereas other samples with a low expression value (i.e., close to the noise level) could contain a reference genotype because the variant peptide cannot be detected in the samples. In contrast, in the case where only the alternative genotype is observed in all 10 samples, the detected variant peptide shows a relatively high expression (i.e., significantly higher than the noise level) in all samples, which we may incorrectly assign a SNP to the 10 samples based on their expression. To avoid this mis-assignment, we use a cutoff value for the minimum intensity of each detected variant peptide (**Panel b**). The higher the cutoff value, the more the variant peptides used (**Panel c**), the more the SNPs derived (**Figure 3d**). For example, the variant peptide #1 (p1) in our example shows a low expression level in 7 samples (grey dots) and a high expression level in 3 samples (black dots), with a minimum level of 12.06 (**Panel d**). We, therefore, can derive a high-confidence SNP from this variant peptide. In contrast, the variant peptide #10 (p10) shows a high level of abundance in all 10 samples, with a minimum level of 18.96 (**Panel e**). It is highly likely that only the variant peptide is detected in all 10 samples. Thus, the SNP derived from this example could be mis-assignment, leading to a low Cscore. Therefore, the higher the cutoff value, the lower the Cscore (**Figure 3d**).

Comment #3: *Supplementary figure 2 and 4 captions are mixed up*

Response: We thank the reviewer for careful reading. We have now corrected this careless mistake.

a

Variant peptide	Chr:position	Reference genotype	Alternative genotype	Samples									
				S1	S2	S3	S4	S5	S6	S7	S8	S9	S10
GISFDAATSGGSASSEK	15:76578762	G	A	13.21	15.92	17.14	12.06	12.61	16.31	13.25	13.17	12.57	16.14
TQLHEDLLPR	7:157959895	C	T	16.28	15.75	13.07	16.34	15.89	14.16	15.22	15.10	15.69	13.61
GIIDLIEER	3:158366900	G	A	14.62	14.99	14.63	14.97	14.31	15.30	13.26	14.95	13.46	15.07
LAEDLGQAEELR	1:21797204	G	C	13.68	17.32	14.22	14.55	16.31	16.38	16.22	15.71	15.38	14.70
HWIKEQEDYIR	4:119444539	C	T	14.41	15.99	14.57	16.14	14.61	15.89	14.28	16.03	14.16	15.85
HLQTMHGTTTHFGK	10:129868686	A	G	14.79	16.52	14.47	15.30	15.99	15.88	15.94	15.78	15.50	15.27
NNAILLR	6:4126379	C	T	15.52	14.73	15.53	17.95	14.90	14.83	15.10	14.69	15.33	15.67
EVIATVVR	2:54135936	T	C	15.76	16.49	16.01	15.91	16.79	16.28	17.00	16.93	15.23	15.46
FFGSLPDSWAR	11:20404613	T	G	16.66	17.42	17.02	15.77	18.02	17.43	18.39	17.69	16.49	16.28
VPSFETAEGIGAEK	4:42020142	A	G	20.65	19.36	20.51	20.03	20.14	19.67	19.56	18.96	20.66	20.16

b

c

d

e

Comment #4: Supplementary figure ??: Strategy of genotype assignment  Rows are shown with index j and are called samples. Are these SNP samples? Maybe they should just be called SNPs? The columns are "Genotypes", are they the samples? But then you select out SNP j and call it Genotype i ? This figure should be labeled more clearly. And maybe replace "Genotype number" with "Number of unique genotypes in sample database". That makes things more transparent.

Response: Thank you again for pointing out this mistake. We have corrected it by removing "SNP j " from Supplementary Fig. 3. The step in this Figure is to show that an example of a selected genotype is used to determine the number of genotypes across all samples.

Comment #5: *Supplementary Figure 6b: How can the number of features be 0, yet the number of matched samples be > 0%?*

Response: The number of features is 6 at the correction coefficient of 0.55. The scale on the left y-axis is relatively large, which makes it look close to 0.

Comment #6: *Line 282: “Qualitative alleles from genomic and transcriptomic data” This is not entirely true. Genomic and transcriptomic allele information comes from stacking reads at a variant locus. For example, in the case of a heterozygous allele, enough reads of the variant need to be stacked up in order for the pipeline (say, GATK) to properly call the variant. So it is a quantitative process.*

Response: We thank the reviewer for careful reading. We have corrected it by removing “Qualitative” in the revised manuscript.

Minor edits:

Comment #7: *Throughout text: “sample mixed-up”  “sample mix-up”*

Response: We have corrected it across the entire manuscript.

Comment #8: *Line 37: “...chromatin sequencing data”  “chromatin sequencing (ATAC-seq) data”*

Response: We have corrected it as suggested.

Comment #9: *Formula between lines 106 and 107:*

$$y_i = (x_i - \min_i(x_i)) / (\max_i(x_i) - \min_i(x_i))$$

Showing that the min and max functions are carried out with respect to the samples i clarifies that you are finding the max and min across samples, and that the scaling is bringing all samples to the same scale.

Response: We have now corrected it in the revised manuscript as follows,

“

$$y_i = \frac{x_i - \min(x)}{\max(x) - \min(x)}$$

Where x is the intensity across all samples, x_i and y_i are raw and scaled intensity for a specific sample, respectively. The scaled intensities are in the range of 0 to 1. ”

Comment #10: Line 137: “exacted”  “extracted”

Response: Corrected.

Comment #11: Line 217: Unnecessary newline character

Response: Corrected.